# A review of wind turbine main-bearings: design, operation, modelling, damage mechanisms and fault detection

Edward Hart[1], Benjamin Clarke[2], Gary Nicholas[2], Abbas Kazemi Amiri[1], James Stirling[1], James Carroll[1], Rob Dwyer-Joyce[2], Alasdair McDonald[1], and Hui Long[2]

[1]Wind Energy & Control Centre, Electronic & Electrical Engineering, University of Strathclyde, Glasgow, UK
[2]Department of Mechanical Engineering, University of Sheffield, Sheffield, UK

**Correspondence:** Edward Hart (edward.hart@strath.ac.uk)

**Abstract.** This paper presents a review of existing theory and practice relating to main-bearings for wind turbines. The main-bearing performs the critical role of supporting the turbine rotor, with replacements typically requiring its complete removal. The operational conditions and loading for wind turbine main-bearings deviate significantly from those of more conventional power plants and other bearings present in the wind turbine power-train, i.e. those in the gearbox and generator. This work seeks to thoroughly document current main-bearing theory in order to allow for appraisal of existing design and analysis practices, while also seeking to form a solid foundation for future research in this area. The most common main-bearing setups are presented along with standards for bearing selection and rating. Typical loads generated by a wind turbine rotor, and subsequently reacted at the main-bearing, are discussed. This is followed by the related tribological theories of lubrication, wear and associated failure mechanisms. Finally, existing techniques for bearing modeling, fault diagnosis and prognosis relevant to the main-bearing are presented.

## 1 Introduction

In 2017 Europe installed a record total of 16.8 GW of additional wind power capacity, bringing the net total installed capacity to 168.7 GW (Wind Europe, 2017). Wind energy is therefore playing a key role in the decarbonisation of the power sector, and so also to the effort to avoid catastrophic climate change. In order to ensure the continued success and growth of the wind industry, the economic viability of wind installations must be maintained. Crucial to this is the reliability of wind turbines and their sub-components, an area which overall has received a lot of attention.

The motivation for this current review is the observation that the wind industry has identified wind turbine main-bearing (WTMB) failures as being a critical issue in terms of increasing wind turbine (WT) reliability and availability. Recently reported figures show that main-bearing (MB) failure rates (over a 20 year lifetime) can be as high as 30% (Hart et al., 2019). Additionally, industry experts at the 2016 WT Drivetrain Reliability Collaborative Workshop (Keller et al., 2016) consistently identified the MB as being the second most important reliability challenge after WT gearboxes in a diverse range of areas, including: MB failure modes, standards and certification, modelling and measurement of internal axial motion, condition monitoring techniques and lubrication. Perhaps surprisingly, given this recent spotlight on MB failures, WTMBs have received

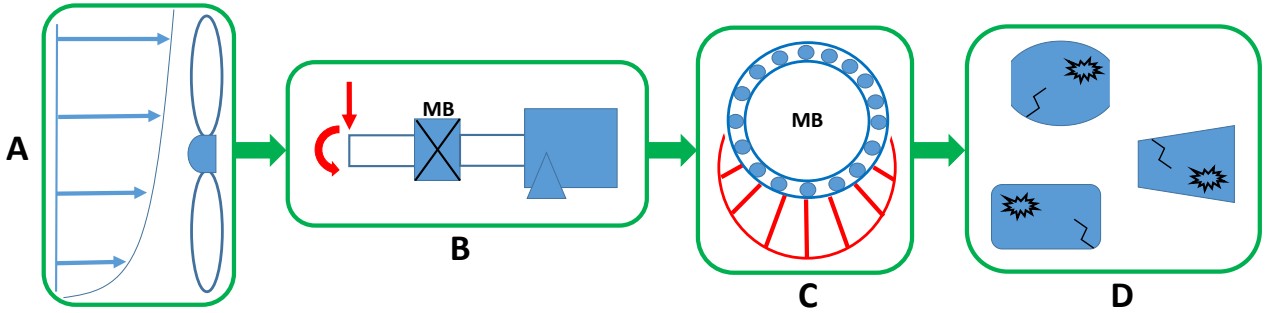

**Figure 1.** Flow diagram capturing the loading relationships which ultimately lead to main-bearing damage and failure.

much less attention in the research literature than other drivetrain components. This is likely due to the fact that historically the MB has not been reported as resulting in high rates of failure. For example, prominent and often cited reliability studies (Carroll et al., 2016; Hahn et al., 2006; Wilkinson et al., 2011; Spinato et al., 2009) either neglect the MB entirely, or appear to lump it in with other components under the heading of 'Drivetrain', 'Main-shaft' or 'Other', obscuring contributions from the MB itself. These works are often simply reporting the information available to them, and so in turn it seems likely that the MB has not been prioritised in terms of logging failures and considering root cause. This might indicate that MB failures are becoming more of a problem as WTs increase in size, however, there is currently no openly available data with which to test such a hypothesis. As suggested in Hart et al. (2019), there therefore seems to be a significant MB knowledge gap which includes failure data, modelling, measurements and design standards. This is in stark contrast to the hundreds of papers in the literature which consider all aspects of WT gearboxes and generators. The current review seeks to aid the closing of this gap by documenting existing literature which considers, or is relevant to, WTMBs. It is intended that this will form a solid foundation which both motivates and supports much needed future work in this area.

The MB is somewhat of an anomaly in terms of WT drivetrain components owing to the fact that it directly interacts with both the rotor, and associated wind-field dynamics and aerodynamically induced loads, and the other rotating components located further down the drivetrain. As such, the MB sits at an interface between various diverse disciplines, including wind field dynamics, WT operation and control, mechanics and loading of rotating machines and tribological failure mechanisms. The relationship between these various factors which contribute to MB lifetime are captured in Figure 1. At **A**, interactions between the wind field and turbine controller generate loads across the rotor which are then reacted by the MB unit, as shown in **B**. This in turn leads to the internal load conditions experienced by bearing rollers and raceways, shown in **C**, the characteristics of which drive tribological mechanisms which ultimately lead to damage and failure of MB components, shown in **D**. This review will therefore proceed in a manner which mirrors the flow of cause and effect seen in Figure 1, outlining both underlying theory and documented results in order to help facilitate cross disciplinary understanding with respect to these various stages.

Section 2 gives a brief overview of wind turbine technology and outlines the cases under consideration in the current work. Section 3 considers wind field structure, rotor loading and operation and control, i.e. aspects of **A** in Figure 1. Section 4 then outlines the most common choices for MB configuration and rolling elements. The modelling of bearing and roller loads, **B**

and **C** in Figure 1, is presented in Section 5. Section 6 details the tribological mechanisms which lead to MB damage and failure, **D** in Figure 1, and then Section 7 outlines the standards which govern MB design and certification. Finally, Section 8 provides an overview of the existing literature on MB fault diagnosis and prognosis.

This paper focuses on rolling element bearings since this is the current technology used in all operational WTMBs. It is worth noting that journal (i.e. sliding) bearings are also being investigated in terms of MB applications (Kasiri et al., 2019; Schröder et al., 2019), but, these systems are still in the R&D phases at present.

## 2   Wind Turbines

While historically there have been various designs for wind turbines, the technology has now standardised to consist almost universally of three bladed horizontal axis machines (Burton et al., 2011). Furthermore, for reasons of improved efficiency and control these machines tend to be variable speed and pitch regulated. The rotational speed is varied when operating in wind speeds corresponding to below rated power in order to maintain optimal aerodynamic efficiency. Once rated power is reached, normally at around 12m/s wind speed, rotational speed is held constant and the blades pitch as wind speeds increase further in order to prevent power and loads exceeding design limits. There is a general trend of upscaling for wind turbine installations, with larger machines appearing each year. This trend is captured in Figure 2 which shows yearly average rotor diameters and power ratings for wind turbines in Germany and the US from 2006 to 2018. Drivetrain choices have proved an area for which a consensus on optimal design has not yet been reached, the main split relevant to the current work being between geared and direct-drive (DD) machines. Geared machines use a gearbox to step up the slow rotational speed of the turbine rotor to the fast (around 1800 rpm) rotational speed needed to generate electricity using conventional generator technology. The gearbox is a complex piece of machinery which has traditionally been seen as a fault prone component that reduces overall reliability. In response to this, DD machines were developed. These remove the gearbox entirely and, in order to be able to generate at slow rotational speeds, have increased generator diameters and operating torques. But, this switch to low speed generation comes with associated increases in cost, size and weight and so the optimal solution is still an open question, especially considering increasing reliabilities of geared machines. This work will consider both drivetrain types for review.

## 3   The Incident Wind Field and Main-Bearing Input Loads

The principal role performed by the MB is that of supporting the rotor while reacting non-torque loads either independently, preventing them being transmitted further down the drivetrain, or in combination with the gearbox and mounts. A thorough understanding of WTMB science requires an appreciation of the loading being reacted at the MB. Therefore, the various components which contribute to MB loads, along with their associated characteristics, are presented and discussed in the current section. Incident loads at the MB are those transmitted from the WT rotor to the hub, and then passed through the cantilevered low-speed shaft (LSS) to the MB itself. Loading across the rotor is in turn determined by the structure of the incident wind field. This characterisation of MB input loads therefore begins with a general discussion of wind field structure.

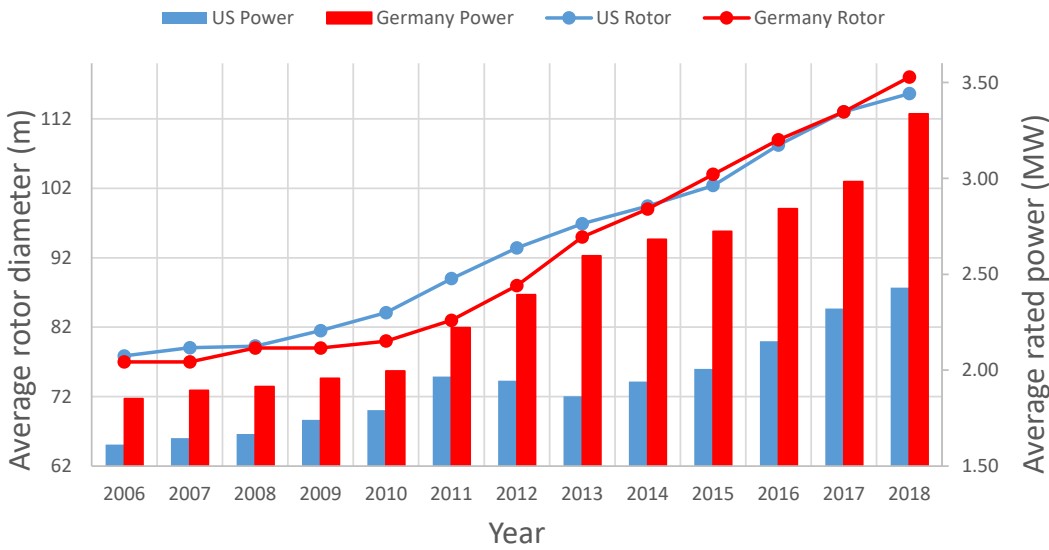

**Figure 2.** Average rotor diameters and rated powers for wind turbines in Germany and the US from 2006 to 2018. Data sources: Fraunhofer (2018) and US Department of Energy (2018)

Hub loads resulting from rotor interactions with the wind field, and subsequently reacted at the MB via the LSS, are then considered.

### 3.1 The incident wind field

Wind turbines generate electricity by interacting with the oncoming wind field. These flow fields tend to be turbulent and complex, containing various structures of differing frequency and magnitude, all of which contribute to the loading imparted to the wind turbine and its components. There are many excellent treatises which cover this topic in detail (Mann, 1998; Brand et al., 2011; Stevens and Meneveau, 2017) and so rather than being exhaustive, the present section seeks only to give a general overview of the aspects of this topic which are most relevant to rotor and, therefore, MB loading.

#### 3.1.1 A homogeneous wind field

Ignoring physical effects, such as ground and structural interactions (these will be discussed below), a wind field can be described as a spatially coherent turbulent flow. Turbulence refers to short timescale variations in local wind speed, usually over about ten minutes or less. In order to give a feel for wind field structure, the most common approaches to their being characterised and modelled will be outlined.

The overall wind field is often interpreted as consisting of turbulent fluctuations superimposed onto slower, larger-scale fluctuations such as those related to changing temperatures or weather patterns (Burton et al., 2011). The most common

measure of turbulence level is *turbulence intensity*, defined as,

$$I = \frac{\sigma_u}{\overline{u}}, \tag{1}$$

where $\overline{u}$ and $\sigma_u$ are the mean and standard deviations of wind speed measurements, usually over a ten minute period. Spatial coherence, in the wind field definition given above, refers to the fact that wind speeds at points separated in space throughout the wind field are not independent of one another. This is due to wind being a viscous medium, along with the fact that some wind field structures can be fairly large, even with respect to the size of modern wind turbine rotors. The turbulent structure of wind field fluctuations at a point are characterised in frequency space by an autospectral density function, $S_u(n)$, for $n$ a frequency in Hz. The two standard models for $S_u(n)$ are the Kaimal and von Karman spectra (Burton et al., 2011). A coherence function then determines the rate at which correlations between turbulent fluctuations decay as a function of separation, $\delta$, in space. For both spectra mentioned above, coherence models, $C(\delta, n)$, generally take the form of a decaying exponential (Burton et al., 2011),

$$C(\delta, n) = \exp\left(-\delta h(n)\right), \tag{2}$$

where $h(n)$ is a monotonically increasing function of $n$, specified differently for each turbulence model realisation. Mann (Mann, 1998) has also developed a turbulence model using a three dimensional tensor representation of turbulent spectra. This model removes the assumption, implicit for the other two, that the turbulence components in horizontal, lateral and vertical directions are independent of one another. This is mentioned for completeness, for the purposes of the current paper it is a general appreciation of turbulent wind field structure which is of principal importance.

### 3.1.2 Physical Interactions

**Wind shear**: In reality a wind field will never be perfectly homogeneous, in large part due to the friction interactions of the wind with the non-smooth surface of the earth. This results in a sheared flow, with wind speeds increasing with height. For neutrally stable wind fields it has been shown that wind shear can be well modelled as (Burton et al., 2011),

$$\overline{u}(z) \propto \ln(z/z_0), \tag{3}$$

at height $z$ and for roughness length $z_0$. Roughness length is a parameter which reflects the size of surface asperities at a given site. Its values range from about 0.001 for a flat desert or rough sea, to 0.7m for a city or forest (Burton et al., 2011) . The relationship is commonly approximated using a power law shear profile,

$$\overline{u}(z) \propto z^{\alpha}. \tag{4}$$

IEC and GL standards (IEC, 2016; GL, 2010, 2012) specify values of $\alpha$ of 0.2 for normal onshore and 0.14 for normal offshore conditions. Although, values of $\alpha$ of up to 0.6 have been recorded (Hart et al., 2016).

**Tower shadow** : Additionally to considering the structure of the wind field itself, it is important to consider the impacts of placing a wind turbine into the air flow. With respect to blade and hub loading on a wind turbine rotor, a significant effect

is due to tower shadow; this being the blocking effect caused by the tower on the wind flow directly in front of the structure (Dolan and Lehn, 2006). The result is a significant drop in wind speed seen by each blade as it passes the tower. This in turn will impact the dynamic loading across the rotor, and so also the loads incident on the MB.

**Yaw error** : Wind turbines can pivot about the tower top in order to ensure the rotor is always facing into the wind. However, it is well known that there are commonly errors in this process, leading to the turbine not facing directly into the wind field (Micallef and Sant, 2016). These yaw errors can result from faulty measurements, bad calibration or simply time lag between the wind changing direction and the turbine following, due to the relatively slow yaw mechanism. From an aerodynamic point of view this results in local wind speeds and blade inflow which change with azimuth angle about the rotor. As with the previous phenomena, this will result in time varying rotor, and so also hub, loads (Micallef and Sant, 2016).

**Turbine wakes** : A wind turbine produces electricity by extracting energy from the incident wind flow. This manifests itself as a slowing of the wind passing through the rotor as kinetic energy is removed from the flow[1]. The resulting velocity deficit behind the turbine then persists as the flow travels further into the wind farm and this is the wind turbine wake. A turbine wake will slowly expand and recover as it moves further through the wind field due to pressure change and turbulent energy exchange from higher in the flow (Stevens and Meneveau, 2017). However, this recovery process will generally only be partial with respect to standard wind turbine array spacing and depends on various aspects of the wind field such as turbulence levels and atmospheric stability (Stevens and Meneveau, 2017). Subsequent rows of turbines in a wind farm will therefore have inflow which includes superimposed wakes from upstream turbines. The dynamics and modelling of wind turbine wakes and the implications for turbine loading and performance is a vast subject which is still very much under investigation (Archer et al., 2018; Adaramola and Krogstad, 2011).

### 3.1.3   More complex wind structures

Other structure can be present in a wind field, either caused by more complex physical interactions or inherent to the wind field itself. The former of these includes complex terrain in close proximity to a wind turbine, for example hills, valleys or forests can all significantly altered the flow regime. Inherent wind field structure can also deviate significantly from the standard representations outlined above. Examples of this include: veered flow (a shear like effect where wind *direction* changes with height rather than wind speed), coherent turbulence phenomena (Kelley et al., 2005) and low-level jets (Gutierrez et al., 2017, 2014). As with wind turbine wakes, the development of theory to model and relate these phenomena to turbine loading and performance is ongoing.

## 3.2   Hub loading

WT hub loading results from a combination of aerodynamic, gravitational and inertial loads on the rotor (Burton et al., 2011), with a significant contributions coming from rotor weight and blade loading. The forces (as opposed to moments) acting on the

---

[1]Technically, at the rotor itself it is in fact a pressure energy which is extracted, however, the net effect relevant to the current work is the resulting velocity deficit.

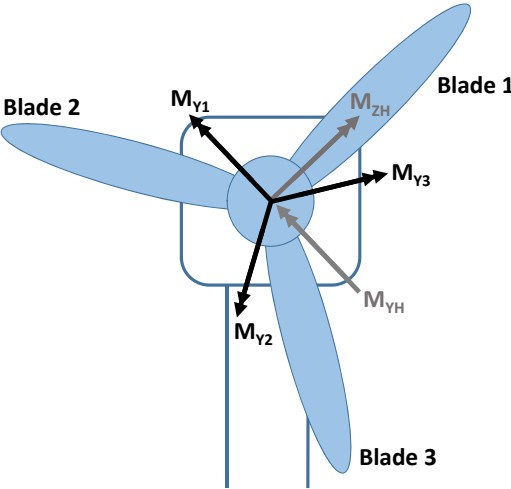

**Figure 3.** Blade-root and hub moments. The blade-root bending moments are in rotating frames referenced to each blade. Hub moments are in a rotating frame referenced to Blade 1.

hub will be dominated by the rotor weight acting vertically and the thrust force in the downwind direction. In addition, each blade is exposed to both in-plane and out-of-plane loading (with respect to the rotor plane), and so generate moments at the blade root in addition to their force contributions. With respect to a rotating reference frame[2] in which each blade root-moment is expressed with axes set respectively perpendicular and parallel to the given blade, see Figure 3, the resultant out-of-plane moments on the hub can be expressed as follows for a three bladed rotor (Burton et al., 2011),

$$M_{Y_H} = M_{Y_1} - \frac{1}{2}(M_{Y_2} + M_{Y_3}),\tag{5}$$

$$M_{Z_H} = \frac{\sqrt{3}}{2}(M_{Y_3} - M_{Y_2}).\tag{6}$$

These equations result from projecting each of the $M_{Y_i}$ onto the relevant hub moment and summing the results[3]. For example, the projection of $M_{Y_2}$ onto $M_{Y_H}$ can be seen to be $-\cos(60°)M_{Y_2} = -\frac{1}{2}M_{Y_2}$. Similarly, the projection of $M_{Y_3}$ onto $M_{Z_H}$ is $\cos(30°)M_{Y_3} = \frac{\sqrt{3}}{2}M_{Y_3}$. The remaining terms contained in the hub moment equations follow along the same lines. Furthermore, rewriting the blade moments as fluctuations about a stationary mean (identical for each blade in a fixed or slowly varying wind profile),

$$M_{Y_i} = \overline{M}_Y + \Delta M_{Y_i},\tag{7}$$

---

[2]The reference frame used here is that of the original source and was kept for the sake of consistency. In general the GL coordinate system (GL, 2010) is more commonly used.

[3]Note that for a three bladed turbine the angle between neighbouring blades is $120°$

observe that inputting these new expressions into Equations 5 and 6 results in the mean values cancelling (since the coefficients of the identical $\overline{M}_Y$ terms will sum to zero), hence the resultant hub moments can be seen to mainly be driven by blade root moment fluctuations about the spatial mean over the rotor circumference.

The aerodynamic blade loads which generate the $M_{Y_i}$ can be decomposed into **deterministic** and **stochastic** components, relating to the physical and turbulent wind field components of Section 3.1 respectively.

### 3.2.1 Deterministic aerodynamic hub loads

Deterministic aerodynamic loads on the rotor result from rotation in a wind field which varies spatially due to shear profile, tower shadow, yaw error *etc*. The resulting load fluctuations on each blade have a frequency which corresponds to the rotational speed of the turbine, $\Omega$. From the point of view of hub loads in a fixed reference frame, a single rotation sees all blades make a full pass through the wind field, and hence the load frequency for the hub is that of the blades increased by a factor of $n$, the number of blades. For a 3 bladed turbine the deterministic component of hub load fluctuations therefore corresponds to $3\Omega$. In this case, deterministic fluctuations in blade root moments, $\Delta M_{Y_i}$, resulting from wind shear and yaw misalignment can often be reasonably well approximated as sinusoids with phase shifts of $0°$, $120°$ and $240°$ respectively (Burton et al., 2011). From Equations 5 and 6 it follows that, in the case of sinusoidal load variations on a 3 bladed turbine, blade root bending moments with range $M_{range}$ result in hub moment fluctuations of $1.5 M_{range}$. Note, the easiest way to see this result is to plot the two functions $f_1(x) = \sin(x) - 1/2(\sin(x + 240°) + \sin(x + 120°))$ and $f_2(x) = \sqrt{3}/2(\sin(x + 240°) - \sin(x))$, both of which vary between 1.5 and -1.5 and so have ranges which are 1.5 times that of $\sin(x)$. The above claim for moment fluctuations is then simply a scaled version of this same example. The very localised variation in blade loads caused by tower shadow will then further increase the load range seen at the hub. Along with $3\Omega$ peaks in the hub load spectra, one would also expect to see associated harmonics of these ($6\Omega$, $9\Omega$, ...), along with a peak at $\Omega$ itself caused by small manufacturing differences between blades. It should be noted that the moment introduced to the turbine shaft from the gravitational force acting at the hub will be opposed by the overturning moment caused by wind shear.

### 3.2.2 Stochastic aerodynamic hub loads

The wind field itself is non-stationary, turbulent and continually evolving. These wind field variations will therefore lead to variations in the turbine hub loads and moments. Taking moments about an axis parallel to one of the three blades, e.g. $M_{Z_H}$, the expression for hub moment variance reduces to a double integration across the other two blades and is identical for both $M_{Z_H}$ and $M_{Y_H}$ (Burton et al., 2011). This moment variance is given by (Burton et al., 2011),

$$\sigma_{M_H}^2 = \sigma_u^2 \left( \frac{1}{2} \rho \Omega \frac{dC_L}{d\alpha} \right)^2 \int\limits_{-R}^{R} \int\limits_{-R}^{R} \rho_u^0(r_1, r_2, 0) c(r_1) c(r_2) \frac{\sqrt{3}}{2} r_1 \frac{\sqrt{3}}{2} r_2 |r_1||r_2| dr_1 dr_2, \tag{8}$$

where $\sigma_u^2$ is wind speed variance, $\frac{dC_L}{d\alpha}$ the blade lift curve slope, $c(r_i)$ the chord length at the given radius of blade $i$ and $\rho_u^0$ the normalised cross-correlation function of the wind field. Crucially, this equation demonstrates the dependence of hub moment fluctuations on turbulence, wind field structure and blade aerodynamic design. As in the deterministic case, the interaction of

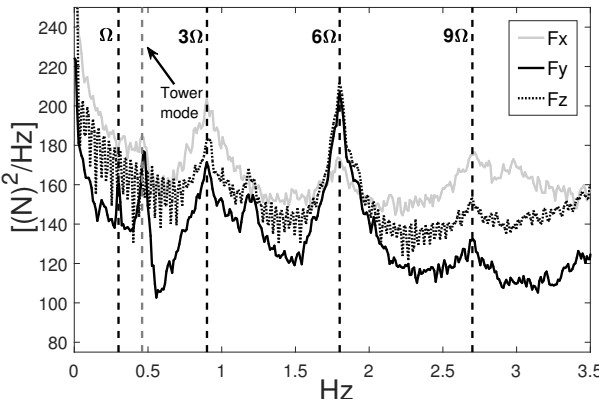

**Figure 4.** Power spectra of WT hub **forces** in the stationary frame. Prominent peaks can be seen to occur at multiples of $3\Omega$.

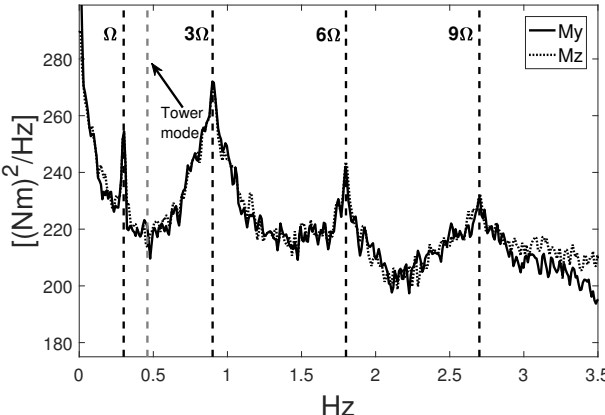

**Figure 5.** Power spectra of WT hub out-of-plane **moments** in the stationary frame. Prominent peaks can be seen to occur at multiples of $3\Omega$.

turbine blades with stochastic features in the evolving wind field result in fluctuating loads at a frequency equal to the rotational speed multiplied by blade number, and then also at associated harmonics.

### 3.2.3 Hub load spectra

In order to illustrate the loads behaviours discussed in the previous sections, example hub load power-spectra (in the stationary frame) are given in Figures 4 and 5. These were generated using data from the simulation of a 3 bladed 2MW wind turbine model in DNV-GL Bladed aeroelastic software. When studying spectra it is important to note that the random nature of stochastic loading will spread the peak out about its central value, hence, sharp peaks stem from deterministic sources and wide peaks from stochastic ones. As discussed, both deterministic and stochastic sources drive loading at multiples of $3\Omega$ for

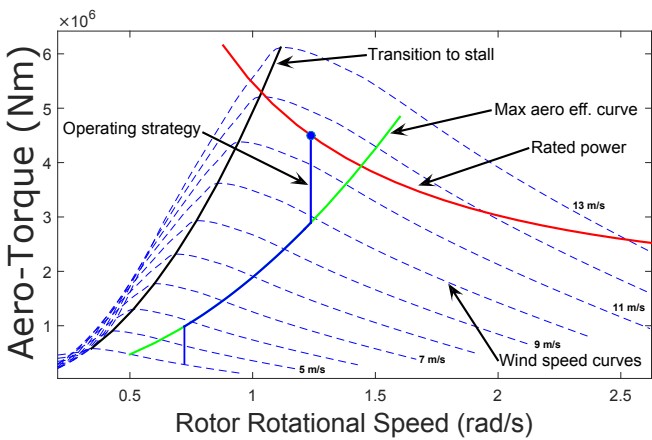

**Figure 6.** Example torque-speed diagram for a 5MW variable-speed pitch-regulated wind turbine model (Hart, 2018). Pitch control becomes active at the point where the operating strategy reaches the rated power curve.

a 3 bladed turbine and so there will be overlap between the different forms of excitation, as can be seen in the figures. Note also some deterministic force loading is present at a frequency which matches the tower natural frequency, and similarly for moments at a frequency that corresponds to rotational speed. Note also that $F_x$ in Figure 4 is thrust force on the hub. Thrust is reacted by the MB along with out-of-plane forces and so a MB experiences continuous variations in both axial and radial loads
at these specific frequencies.

### 3.3   Turbine operation and control

While wind turbine power generation is commonly discussed in terms of above-rated and below-rated conditions, from an operational standpoint each case breaks down into further regions with distinct operational requirements. Figure 6 shows an example design trajectory of a variable-speed pitch-regulated wind turbine as a function of torque and rotational speed. The
optimal efficiency tracking region and initial point of pitching can be seen, along with constant speed transition regions. This diagram highlights the fact that there are various regions of wind turbine operation with different design trajectories and associated wind speeds, hence, the load characteristics for the MB will also be different in each case. An additional operating case not seen here is the shutting down of a turbine in above rated conditions when wind speeds consistently stay above a design limit of around 25m/s. Wind turbine operation can additionally include measures to meet certain performance or grid
support standards, such as power curtailment or emergency stops. For a detailed discussion of turbine operation and control regions see Hart (2018).

Having outlined various regions/types of wind turbine operation, it is important to note that achieving these operational requirements is then a control task. Like in other industrial applications, wind turbine controllers measure how close to the design trajectory the machine is at a given time and, in cases of deviation, take control action to remove any error. For example,
in above rated pitch operation the power is measured and the blades pitched in the appropriate direction if the power is found

to be either above or below its rated value. There are many types of wind turbine controller (Novaes Menezes et al., 2018) and each turbine must have a controller tuned to its specific dynamics. As well as following a design trajectory or curtailing the turbine, modern controllers are increasingly required to perform additional tasks such as load alleviation for blades and tower, the provision of grid frequency support and damping of system resonances (Hart, 2018). These various tasks can be achieved using a range of advanced control techniques including the Power Adjusting Controller (PAC) (Stock, 2015), Individual Pitch Control (IPC) and Individual Blade Control (IBC) (Bossanyi, 2003; Han and Leithead, 2015) among others. Each of these control approaches will alter the behaviour of the turbine with respect to an incident wind field and so in turn will impact the loads seen at the hub and MB. To date there have been no studies which evaluate the impacts of different operation and control approaches to MB loads beyond considering stops and emergency stops (Scott et al., 2012).

As can be seen in Figure 6, the LSS rotational speeds seen by an operating wind turbine are slow, with specific ranges being dependent on size and other design considerations. It is also worth noting that rotational speeds tend to decrease as turbine size increases, this is in order to preserve the optimal ratio between blade tip speed and the incoming wind speed.

## 4  Main-Bearing Configurations and Rolling Elements

Wind turbine drivetrains have a number of possible configurations depending on various factors, the most prominent of these being whether the turbine ultilises a gearbox or not. In addition to this consideration, there are loading and cost considerations which drive layout decisions, although of the standard designs there is currently no consensus as to whether one or other is optimal in all cases.

Decisions relating to the use, or not, of a gearbox in the wind turbine largely come down to financial factors and trade offs between capital cost, weight, reliability, maintainability and the OEM's existing expertise. Historically a majority of variable speed wind turbines have used doubly-fed induction generator technology with a multistage gearbox (Polinder et al., 2005). Gearboxes, however, manifest relatively high failure rates and so lost revenue due to down-times and repairs has become a major concern of the wind energy industry (Su et al., 2017; Dabrowski and Natarajan, 2015). New systems have therefore been developed which require gearboxes with fewer stages or are direct-drive (DD), removing the gearbox completely. DD systems are generally more expensive in terms of capital cost due to having large associated dimensions, weight and generators with high torque ratings. However, with no gearbox the component with the highest failure rate has been removed and so these turbines tend to be more reliable than their geared counterparts. As mentioned above, no consensus yet exists as to which generator type is optimal, notably Siemens-Gamesa have committed exclusively to DD technology for all their European offshore turbines in future, so too have GE with their offshore offerings. Whereas Vestas are going with medium and low speed gearbox technology.

### 4.1  Geared turbines

The most common drivetrain configurations for geared turbines are those with *three-point* or *four-point* suspensions, referring to turbines with either a single (SMB) or double (DMB) main-bearing setup respectively (Guo et al., 2016). Examples of these

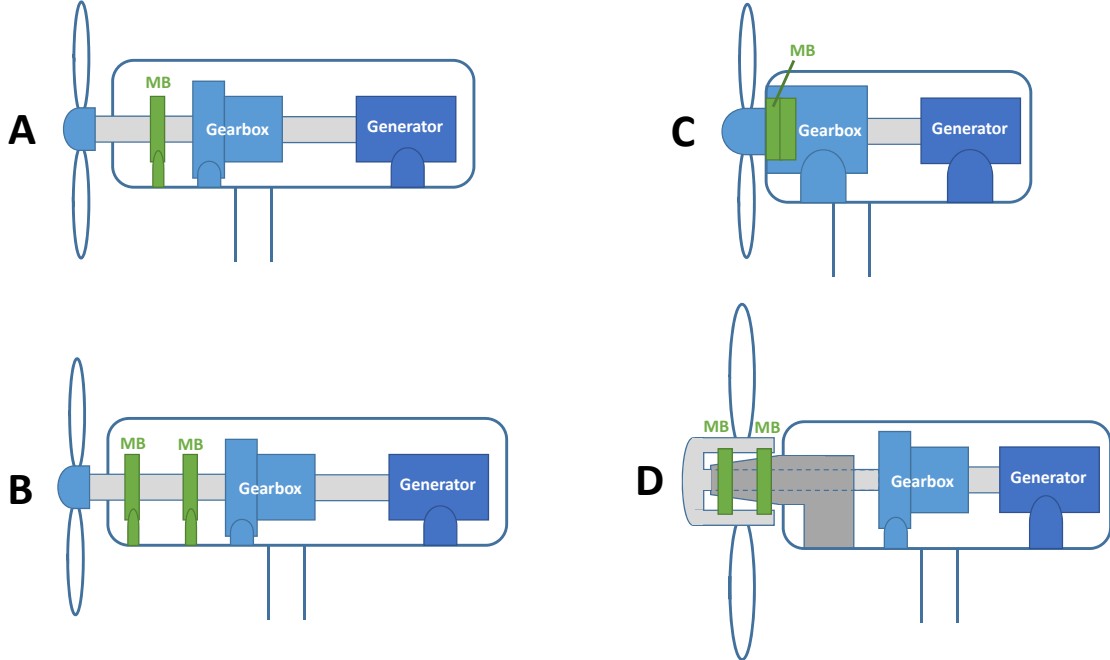

**Figure 7.** Depictions of the existing drivetrain layouts for geared turbines. In alphabetical order these are: single main-bearing, double main-bearing, gearbox integrated main-bearing and a 'floating drivetrain' design respectively.

two configurations are shown in Figure 7, diagrams **A** and **B** respectively. A SMB layout requires gearbox trunnions to react loads at the gearbox end of the LSS and so in combination there are three points of support. This setup has been used in commercial wind turbines by GE, Siemens (prior to becoming Siemens-Gamesa), Nordex and Vestas for machines rated at between 1.5 and 3 MW (Guo et al., 2016). For the DMB layout, a second bearing is placed near the gearbox end of the LSS in an attempt to react non-torque loads before they reach the gearbox. This latter setup has been used in commercial wind turbines by Gamesa (prior to becoming Siemens-Gamesa), Vestas and GE for machines rated at between 2 and 2.5 MW (Guo et al., 2016). DMB designs protect the gearbox more fully from non-torque loads, however, they can be sensitive to misalignment, require more space and have a higher associated capital cost (Bergua et al., 2014; Guo et al., 2016).

Designs have also been implemented which integrate the MB directly into the gearbox, removing the need for a main-shaft and so resulting in a very compact overall design. Such a design is shown in Figure 7-**C**. The benefits of such an arrangement lie mainly in the resulting reduction in tower top weight, allowing for cost savings in the WT tower and substructures (Terrell et al., 2012). However, drawbacks have also been identified including: possible early failures due to incompatibilities between the gearbox and remaining nacelle components (to which the gearbox is now a main load path) (Terrell et al., 2012), noise issues from vibro-acoustic propagation, the potential for poor load distributions across gear faces (Bergua et al., 2014) and, perhaps most importantly, the fact that a MB failure in the fully integrated design case requires the entire nacelle to be exchanged. This

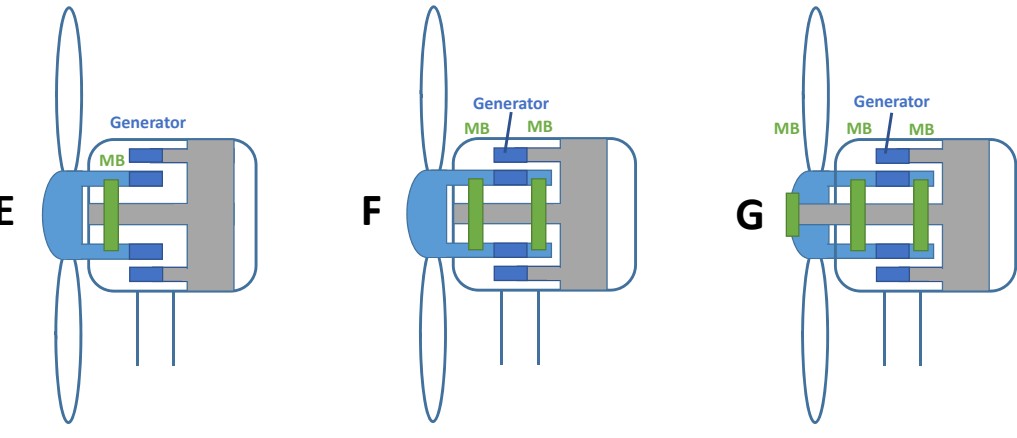

**Figure 8.** Depictions of some of the existing drivetrain layouts for direct-drive turbines. One example is given for each of single, double and triple MB designs.

drivetrain configuration has been used by both Vestas and Areva in 3 and 5 MW wind turbines respectively (Bergua et al., 2014).

Another concept looks to decouple support structure and drivetrain through the implementation of a 'floating drivetrain' design. Two MBs support the rotor as it rotates about a structure fixed directly to the turbine bedplate (Bergua et al., 2014).

Torque is then transmitted to the shaft via an elastic coupling towards the front of the turbine hub, as in Figure 7-**D**. The principal drivers for this design are the removal of non-torque loads from being propagated into the gearbox and generator, however, load improvements for the MBs themselves are also claimed to result from a more even distribution of loads due to the rotor centre of gravity now presiding between the bearings (Bergua et al., 2014). While this may be the case, no published studies exist which specifically investigate resulting loads on the MBs themselves from such a design although, as will be discussed,

this is true in general for all drivetrain layouts. Published figures do show low gearbox failure rates for this drivetrain design (Bergua et al., 2014), but similar figures are not available for the MBs. This design has been used in Alstom turbines for a range of power ratings and will also be used in GE offshore DD installations, as discussed below.

### 4.2 Direct-drive turbines

Current DD wind turbines are almost all radial flux machines, although axial flux topologies do also exist (Dubois et al., 2000).

This work therefore considers the more common radial flux design. Given the integration of support structure and generator in DD turbines, a main design aim becomes ensuring that the specified minimum air-gap clearance is maintained (Stander et al., 2012). This precision task is therefore additionally required of the MBs, along with the transfer of torque and reaction of non-torque loads. Bearings can be arranged in single, double or triple configurations, with various possible layouts in each

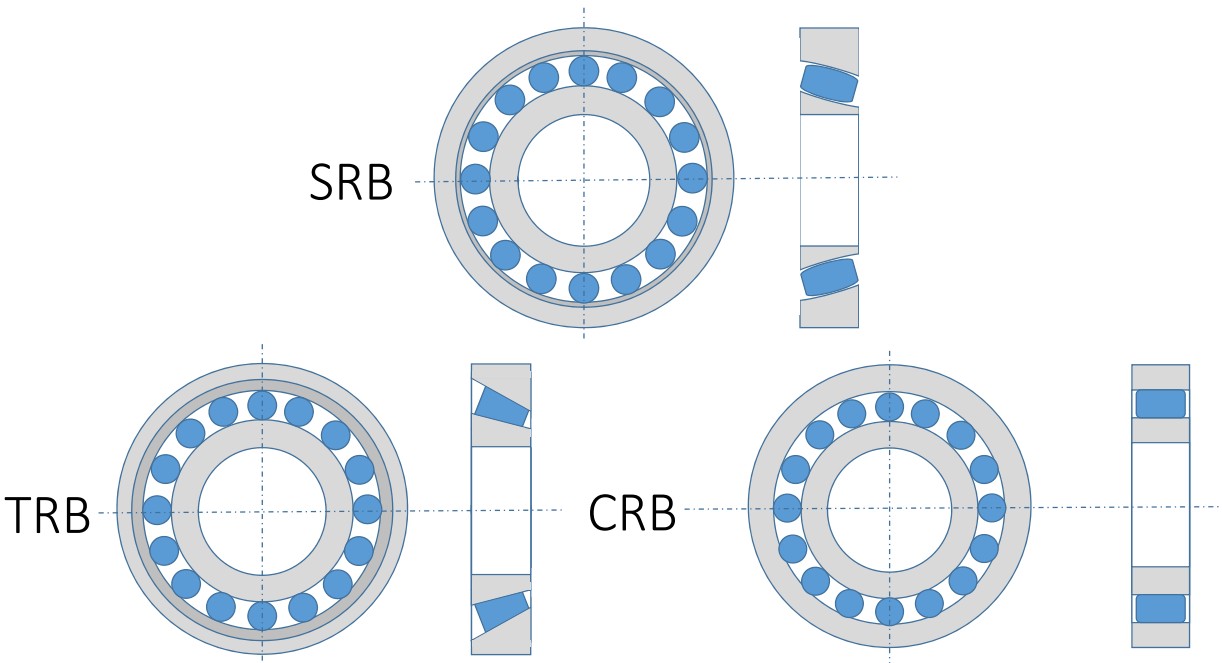

**Figure 9.** Bearing rolling element types. A single row only is shown for each. Note that for SRBs and TRBs, in WTMB applications, a double row back-to-back configuration will generally be used on the rotor side.

case (Stander et al., 2012), including having bearings in the air-gap clearance. Figure 8 shows one example of each, for a full accounting see Stander et al. (2012). The 'floating' drivetrain concept introduced for geared turbines can also be applied in DD cases and is currently being used in GE's 6MW offshore wind turbine.

Maintainability was mentioned above as being a key driver for drivetrain selection at the design stage. Section 4.1 contained a clear example of this with respect to fully integrated drivetrains requiring the entire nacelle to be exchanged in the case of a main bearing failure. This is a more extreme example but it highlights the point that the maintainability of a given system will play an important role in determining its overall viability. From a maintenance point of view, more modular concepts tend to allow for easier component accessibility and exchange. Partially-integrated designs have also been introduced as a way of trying to secure the benefits of both design philosophies (Gasch and Twele, 2012).

## 4.3 Rolling elements

The current section gives an overview of the bearing rolling elements used in WTMBs for the various configurations of the previous section. These rolling elements are depicted in Figure 9.

**Spherical roller bearings (SRBs)**: SRBs are characterised by their outer raceway being a portion of a sphere. The rollers, in turn, are shaped such that they conform closely to both inner and outer raceways. This results in a bearing which is internally

self-aligning and has a high radial load carrying capacity. A double-row SRB (DSRB) configuration, as in the figure, allows for the reaction of combined axial and radial loading. SRBs cannot support moment loads in either single or double-row configurations (Harris and Kotzalas, 2007).

**Tapered roller bearings (TRBs)**: A single row TRB can carry a combination of axial and radial loads. Differences in contact angles at the inner and outer raceways result in an unbalanced force which will act to drive the roller against its guide flange. The bearing contact angle is determined by the magnitude of axial loading to be reacted, relative to radial loading. As with SRBs, TRBs are commonly put into double-row (or more) configurations (DTRB) in order to increase their radial load carrying capacity. TRBs and DTRBs can support moment loads (Harris and Kotzalas, 2007). Numerical modelling has shown the stiffness and displacement of rollers to be sensitive to the magnitude of applied moment and radial loads (Tong and Hong, 2014).

**Cylindrical roller bearings (CRBs)**: CRBs have cylindrical rollers, usually crowned at the ends to prevent high edge stresses. They exhibit low-friction operation compared to SRBs and TRBs and have a high radial load capacity, however, they are unable to support axial loads. Tolerance to small axial loads can be achieved by incorporating a thrust flange into the bearing housing. As previously, a double row (DCRB) may be be used to increase load carrying capacity (Harris and Kotzalas, 2007).

**Toroidal roller bearings (TorRBs)**: Similar to SRBs, but with a toroidal outer raceway and elongated rollers by comparison, TorRBs offer an increased radial load carrying capacity while also retaining some of the self-aligning properties of SRBs. TorRBs can only support small axial loads.

Having outlined common roller types and their typical configurations, we will now detail which bearings are used in the various drivetrain setups. Two important concepts necessary for this discussion are **locating/non-locating** bearings and **preloading**.

When two or more bearing are being used to support a shaft, in order to prevent excessive axial motion one or both bearings must be fixed in the axial direction. However, changes in temperature during operation (either machine related or due to external influences) will result in thermal expansions which, in the case of two axially fixed bearings, would lead to material stress and deformation. It is therefore generally necessary to have one bearing which is free in the axial direction and so able to accommodate axial displacements (the non-locating bearing), and one which is axially fixed in order to hold the shaft in place (the locating bearing).

Radial preloading of roller bearings is used to increase the number of rolling elements under load around the bearing circumference. This has the effect of reducing the maximum rolling element load and helps prevent skidding[4] (Harris and Kotzalas, 2007).

**Geared turbines**: Geared turbines which have either a SMB or DMB setup have historically most commonly used a locating DSRB for the rotor side bearing, along with a SRB, CRB or DTRB as the generator side bearing (Yagi, 2004). The use of SRBs has remained prevalent, along with the introduction of designs featuring two TRBs or a DTRB and CRB (at rotor side and generator side respectively) in the DMB drivetrain case (Yagi and Ninoyu, 2008). More recently, SMB designs which

---

[4]A combination of rolling and sliding.

replace the rotor side DSRB with a pre-loaded DTRB or triple-CRB have become available (Fierro, 2017; Schaeffler; Liebherr). Finally, a design also exists for the DMB case which uses a non-locating TorRB and locating DSRB combination (SKF). In the case where the MB is integrated into the turbine gearbox both DTRB and triple-CRB setups have been used (Yagi, 2004; Bergua et al., 2014). The 'floating' drivetrain design uses a spread pair of TRBs for rotor support (Bergua et al., 2014).

5. **Direct-drive turbines**: DD turbine mechanical MBs tend to use TRBs, CRBs or TorRBs for rolling elements in either single or double-row formations at each support point of the MB system (Stander et al., 2012). In the case of a SMB with rollers located within the air-gap, disk shaped rollers can be used which are supported by a shaft and small mechanical bearings (Engstrom and Lindgren, 2007).

## 5 Main Bearing Modeling

As with other aspects of MB theory, there are not huge numbers of studies which deal specifically with the modelling of WTMBs. However, there is a substantial literature concerned with the modelling of rolling bearings of different types and under various conditions. This section will therefore outline both the existing modelling work concerned with WTMBs, along with relevant contributions from the wider theory which seem likely to play a role in future MB research.

### 5.1 Hertzian elastic contact theory

Many rolling bearing models are underpinned by Hertz's classical theory of local stress and deformation resulting from two elastic bodies in point or line contact. An excellent overview of this theory can be found in Harris and Kotzalas (2007). The following general load-deflection relationship is obtained from the classical theory,

$$Q = K\delta^n, \tag{9}$$

which relates normal contact force, $Q$, with the radial deflection, $\delta$, at the contact. For a point contact $n = 3/2$ and for a line contact $n = 10/9$. The constant factor, $K$, represents contact stiffness, and is determined from material properties and the geometry of the contact patch which forms between the bodies. Various analytical approximations have been developed (Antoine et al., 2006; Brewe and Hamrock, 1977) which allow for fast evaluation of the terms necessary for determining contact geometry and $K$ itself with minimal error. Tangential forces due to friction and lubrication conditions are then given by,

$$Q_t = \mu Q, \tag{10}$$

where $\mu$ is an equivalent friction coefficient. In the case of roller bearings, the surface will often not be a perfect cylinder and so assuming a line contact can lead to inaccuracies. In order to avoid these errors it has been proposed that the bearing should be partitioned into a number of *slices*, on each of which a perfect line contact is assumed and the forces evaluated as above. The resultant contact force on the entire roller is then a summation, or integration, of those on the slices.

For an applied load resulting in deflection, the point of maximum rolling element load coincides with the point of maximum deflection. Loading at an angle of $\psi$ from the maximum load, $Q_{\max}$, can be evaluated as,

$$Q_\psi = Q_{\max} \left[ 1 - \frac{1}{2\epsilon}(1 - \cos\psi) \right]^n . \tag{11}$$

The parameter $\epsilon$ is related to clearance in the bearing and the radial shift at the centre of the shaft. Its value can be interpreted
as giving the proportion of the total bearing circumference which is under load (Harris and Kotzalas, 2007). Figure 1-**C** shows an example radial load distribution for a rolling element bearing with $\epsilon \approx 0.5$. Similar formulations hold for cases of combined axial and radial loading.

## 5.2   Multibody models

Multibody formulations, modelling the bearing system as a collection of interacting bodies, are most commonly used for
bearing models due to their speed and relative simplicity. These models can be quasi-static or dynamic in nature. In both cases, force expressions are determined which describe loading-deflection relationships between the various bearing components (rollers, races, flanges etc.), with contact loads generally evaluated using the Hertz theory of the previous section. Bearing internal loads and deflections are iteratively solved for, using Newton-Raphson or similar, such that external and bearing forces balance. For the quasi-static case this gives the desired result at each timestep. In the dynamic case, load and stiffness properties
from this first stage can be used in a dynamic model to analyze rolling element motion.

     Ghalamchi et al. (2013) present a general purpose dynamic SRB model which follows the approach outlined above. Computational efficiency is improved by assuming that: sliding does not occur between bearing components, all rollers move around the raceway with equal velocity and don't interact with each other, isothermal conditions exist throughout. Centrifugal forces acting on the rollers are also neglected. Results from dynamic simulations of both single and double bearing configurations
are presented. Both Tong and Hong (2014) and Zheng et al. (2018) develop models to evaluate loading for TRBs subject to combined force and moment loads. The first of these improved upon existing techniques in terms of evaluating pressure distributions along rollers via integration. The latter is specifically considering TRBs for DD floating wind turbine main-shafts and additionally includes effects related to angular misalignment and friction, although cage influences and skew effects are still neglected. Both models are quasi-static in nature. In Jain and Hunt (2011) a dynamic model is derived with which the presence
of bearing sliding or skidding (instead of rolling) can be investigated. The model itself is developed for ball-bearings and so would need to be adapted before it can be applied in the MB case, however, given that sliding is associated with some potential MB damage modes (see Section 6) this would seem to be a key point of investigation for future MB research.

     The multibody models presented here are all fairly general, mainly differing in terms of considered bearing type and assumptions made during the modelling process. The fundamental building blocks are therefore present for further development
of models with which to investigate WTMB loading for different bearing types and configurations. During this development it will be important to appraise the assumptions of the given approaches in order to ensure that the important factors in terms of MB operational loads are accounted for, including a consideration of whether dynamic or quasi-static models are required.

Both SKF and Schaeffler have developed their own multibody simulation softwares for design, testing and optimisation. SKF's software, BEAST (BEAring Simulation Toolbox), can evaluate internal motions and forces in a bearing under specified loads. The model employs an EHL lubrication model and can predict the occurence of phenomena such as wear and fatigue. Schaeffler have developed BEARINX which offers similar features and looks to account for lubricant condition and contamination effects in the model. Similarly, Timken have their own proprietary analysis system, Syber, which they use for design optimisation and component selection. A commercially available multibody modelling software, SIMPACK, has been used to analyse MB loads at a number of operating points, while also exploring the effects of bearing clearance (Sethuraman et al., 2015). In Cardaun et al. (2019) a SIMPACK model is used to investigate the impacts of yaw misalignments on WTMB loads in terms of fatigue life. The results indicate that misalignment increases the damage equivalent loads for the MB, but that the effect is not symmetrical and depends on the inflow direction.

In Hart et al. (2019) the loading experienced by WTMBs is investigated using simple beam-support representations of the drivetrain[5]. Hub-loadings taken from aeroelastic code are used as inputs to models which represent three and four-point mount systems in order to compare the loads seen by each. The results of this work indicate that radial loads seen by four-point mount systems are generally higher than for the three-point system, but, they also saw a corresponding reduction in axial-to-radial load ratios, a quantity for which high values have been associated with potentially damaging events such as unseating of rollers (Hart et al., 2019). Given the reported higher instances of failures in three-point systems in this same work, it is suggested that specific loading combinations (in terms of axial and radial loads), rather than just individual load magnitudes may be important to understanding failures.

### 5.3 Finite element models

Finite Element Methods (FEMs) can be used for modelling of the deformation pressure within a bearing. A function is chosen to represent uniquely the displacement within each node. Then, the element stiffness matrix is obtained from equilibrium and upon obtaining the matrix for the complete bearing, the full matrix is assembled and boundary conditions applied. The solution of the resulting matrix equation produces the nodal displacements (Harris and Kotzalas, 2007). Accuracy of solutions is highly dependent on the number of nodes and so considerable computational power and time is generally required for good results. Due to its extensive computational time and power required, FEM has mainly been used in bearing applications for modelling of roller-raceway contacts (Jackson and Green, 2005; Zeid and Padovan, 1981). There are however also cases of full bearings being modelled using these methods. For roller-raceway contact modelling, calculations of the distribution and magnitude of surface stresses can be carried out which include the effects of roller and raceway crowning (Harris and Kotzalas, 2007). Results are typically compared with Hertzian calculations for validation. Full bearing modelling has been done to analyse fatigue failure in wind turbine gearbox bearings (Grujicic et al., 2016; Jiang et al., 2015; Lai and Stadler, 2016) and in one case for a WTMB under steady state loads (Liang et al., 2013), and also to understand contact loads within a pitch or yaw bearings (Chen and Wen, 2012). More recently, FE models have been used to understand load and pressure distributions about WTMBs. In Kock et al. (2019) this is done with the aim of understanding the effect contributed by elasticity in the MB housing along

---

[5]These technically still fall under the heading of multibody, although they are at the simpler end of this spectrum.

with the applied load and bearing clearance. Their results indicate that the elastic surroundings lead to more rollers under load and hence a lower maximum load, indicating that housing elasticity should be modelled to avoid errors. With respect to the applied load it was found that loading directions are important since the bearing mounts result in variations in elasticity about the bearing circumference. Finally, increases in clearance were found to result in increased bearing loads as fewer rollers are

supporting the system. In Reisch et al. (2018) modelling capabilities were investigated by comparing results from an FE model of a 3MW wind turbine nacelle (including MB), with those coming from an experimental test-bench research system of the same power rating. In the model, bearing rolling elements are modelled by nonlinear spring elements. With respect to the MB the investigations consider interactions between the MB and gearbox planet carrier. Good agreements between modelled and measured values are found in this case.

There exist various forms of commercially-available FEM software. Particularly relevant to the wind turbine case is Ro-maxWIND, developed by Romax Technology and certified by DNV-GL for gear and shaft design analysis.

## 6    Tribological Theory for Wind Turbine Main-Bearings

Tribology is the science of interacting surfaces in relative motion. As such, it is this discipline which is concerned with the metallurgical effects of roller and raceway interactions in rolling bearings, and subsequent damage and failure mechanisms.

The current section seeks to outline the aspects of this theory most relevant to WTMBs.

### 6.1    Lubrication

Lubrication is the separation and reduction of contact pressure of two sliding or rolling contacts resulting from entrapment of a liquid (or grease) into a converging gap at the contacts (Halme and Andersson, 2010). Separation stems from a hydrodynamic pressure and lubricant film forming between the contacts. The ratio of oil film thickness to local roughness on contacts is

denoted $\lambda$ and determines the mechanism of lubrication. When $\lambda < 1$ (Boundary lubrication) the hydrodynamic action is not strong enough to separate the contact surfaces and loads are mainly carried via solid-solid contact, resulting in increased friction and wear (Halme and Andersson, 2010). Various factors effect the $\lambda$ value in practise including the availability of oil, bearing surface roughness, elastic deformation of the loaded surfaces and the viscosity and pressure-viscosity relationship of oils. Elastohydrodynamic lubrication (EHL) is a lubrication condition model which takes these various factors into account.

In EHL the oil film thickness is slightly higher than the overall contact surface roughness, and so $\lambda > 1$. Importantly, assumed conditions of EHL underpin most bearing life calculations (Halme and Andersson, 2010). Mixed lubrication occurs when $\lambda \approx 1$ and a combination of boundary and EHL regimes are present. When $\lambda \gg 1$ the sliding surfaces are completely separated by a fluid film which is thick in comparison to surface roughness. These conditions only generally occur at less heavily loaded contact interfaces, such as those between a roller and its cage or the roller end and bearing race flange (Halme and Andersson,

2010). For WTMBs the specific lubrication regimes at these traditionally moderately loaded contacts have not been investigated directly and hence it is possible that there is divergence from the standard case.

Rolling elements influence the lubricating film layer as they pass. Roller motion ejects oil from the contact area, as does centrifugal action in higher speed applications. Starvation can occur if the oil reflow, back into the raceway, is too slow to replenish the quantity of oil necessary for EHL before the next roller passes (Halme and Andersson, 2010). If a liquid lubricant is used there needs to be some means of ensuring oil flow to contact regions. This can be done through pumping or circulatory splash systems. Grease lubricated bearings are more likely to experience starved conditions as grease is squeezed out of contact regions during roller passage (Halme and Andersson, 2010).

**Oil vs Grease Lubrication**: Oil lubricants are typically employed for high speed and high temperature applications. They require a system for circulation and, therefore, have higher maintenance demands. However, they are known to work better at cooling the lubricated surfaces. Grease, on the other hand, is thickened oil. Its action allows localization of the lubricant to regions of contact within the bearing. It consists of a suspension of fluid dispersed into a soap or non-soap thickener with the addition of a variety of performance-enhancing additives (Harris and Kotzalas, 2007). Grease is typically used in low speed, low temperature applications. It requires less maintenance due to not requiring a circulation system, and is typically the lubricant used in WTMBs. Pumping systems are now employed for grease re-lubrication, removing the need for manual re-greasing by an engineer up-tower.

Grease has a more complex, two phase lubrication mechanism, namely the churning and bleeding phases (Lugt, 2016). The churning phase starts when the bearing is lubricated with fresh grease and is distinguished by macroscopic grease flow. After a period of operation ($\sim$24 hours), a large portion of the grease will have been swept to the side of the rolling region. During the churning phase there is a good supply of lubricant and the contacts tend to remain flooded. After the churning phase, macroscopic grease flow stops and the supply of lubricant into contacts takes place through bleeding flow as the grease releases oil through phase separation (Lugt, 2016). When the moving parts of a bearing come into contact with grease, a small quantity of thickened oil will adhere to the bearing surfaces. The oil is then gradually lost through oxidation or evaporation and with time the oil in grease within the contact region will be depleted. The grease lubricant regime is dependant on the balance of lubricant supply and loss mechanisms which determine the film thickness.

## 6.2 Damage and wear mechanisms

This section details the wear and damage mechanisms most likely to effect WTMBs.

### 6.2.1 Classical fatigue

Dynamic contact conditions generate stress fields and plastic deformation in contacting materials, eventually leading to the formation of fatigue cracks at defects or inclusions in the material structure (Halme and Andersson, 2010). Intersections of such cracks lead to the formation of pits and the releasing of abrasive particles into the bearing environment. Once initiated this process will self propagate through surface roughening, reduced contact areas and increasing levels of abrasive wear. Roller bearing lifetimes, with respect to classical fatigue, can be described statistically as a function of bearing design, load, speed

and lubrication conditions (Halme and Andersson, 2010). The basic formula for the life-rating of a rolling bearing is,

$$L_{10} = \left( \frac{C}{P} \right)^p, \tag{12}$$

where $L_{10}$ is in millions of revolutions, $C$ is the basic dynamic load rating, $P$ the bearing equivalent load and $p = 10/3$ for roller bearings. $L_{10}$ gives the number of revolutions that 90% of a bearing population is expected to survive without experiencing fatigue pit formation (Halme and Andersson, 2010). It is standard practise to modify this basic rating via the multiplication of scalars which account for other factors including speed, oil viscosity, oil contamination or a desired percentile other than 90%. Hence the modified life, $L_{nm}$, is used which is proportional to $L_{10}$,

$$L_{nm} \propto L_{10}. \tag{13}$$

Implicit to rolling bearing fatigue life prediction, using the above formulations, are certain assumptions which it is important to be aware of. Those most pertinent to the current work are as follows (Halme and Andersson, 2010):

- EHL conditions are assumed to hold throughout the bearing lifetime

- Present theory does not account for intermittent operation, most of the lifetime of a rolling bearing is assumed to occur under steady-state conditions

- These equations do not account for the presence of wear instead of, or as well as, rolling contact fatigue.

### 6.2.2 Micro-pitting

Micro-pitting wear has been found to be present in documented cases of WTMB failures to date (Kotzalas and Doll, 2010). Micro-pitting results from an insufficient local lubricant film thickness (potentially resulting from excessive loading or off-design operation) allowing interactions between roller and race surface asperities. In these cases the normal stresses, *i.e.* those accounted for in classical fatigue analysis, are compounded by additional frictional shear stresses. Bulk contact stresses are therefore increased and points of maximum stress move closer to the surface, leading to significant localised stresses beneath asperity contacts and the subsequent formation of micro-pits (Kotzalas and Doll, 2010). Surface asperity interactions associated with this failure mechanism are generally found to occur in situations where there is relative sliding between the contacting surfaces, in addition to the presence of an insufficient lubricant film (Kotzalas and Doll, 2010).

### 6.2.3 Spalling

Spalling is the pitting or flaking away of bearing material, often as a result of some other primary damage mechanism (Timken, 2011). Spalling can occur due to geometric stress concentrations caused by misalignment or excessive loading and by high localised stresses resulting from surface dents and damage or hard particle contamination. Micro-pitting will commonly lead to spalling damage of the effected surface (Kotzalas and Doll, 2010).

### 6.2.4 Smearing

Smearing is a form of adhesive wear which occurs under sliding contact between two surfaces, involving the transfer of material from one surface to the other. In rolling element bearings this is known to occur in cases where roller rotational speeds change rapidly, for example when a roller accelerates on entering a more highly loaded bearing region (Evans et al., 2013b).

### 6.2.5 Abrasive wear and debris damage

The entrainment of hard particles into bearing contacts can lead to physical damage; either from indentations left by rolling particles, or surface scratching due to sliding particles (Nilsson et al., 2006). Similarly, particles in the lubricant, through contamination or abrasive damage of bearing surfaces, can lead to high local stress fields and abrasive conditions which can be worse than those seen in boundary lubricated regimes. Ductile metallic debris in the lubricant can be rolled over and flattened by rolling elements, leaving larger shallow smooth dents. Brittle materials, such as sand, fractures when rolled over and leaves many small but steep sided dents. Debris damage of these types can lead to fatigue crack formation and spalling (Nelias and Ville, 1999; Ai, 2001). The presence of more brittle particulate contamination can lead to abrasive wear and gradual increases in bearing clearance (Nilsson et al., 2006; Dwyer-Joyce, 1999).

### 6.2.6 Fretting

Fretting corrosion occurs at interfaces where loads are transferred under oscillating contact micromovements (ISO, 2017b), commonly a result of vibrations in the system. A WTMB may be exposed to both blade and gearbox vibrations and so this type of damage is possible (Yagi, 2004). Prevention of this type of damage involves trying to optimise bearing clearances and lubricant type.

### 6.3 Damage related to materials and manufacturing

Inclusions within bearing materials are known to increase the risk of premature failures due to them acting as a nucleation site for subsurface cracks and spalling (Jalalahmadi et al., 2011; Lund, 2010; Evans et al., 2013a), much effort has therefore been made to refine steel-making process in order to ensure a high level of bearing material homogeneity and cleanliness (Tabatabaei et al., 2018; Ni et al., 2017; Michelic et al., 2011). As a consequence, the probability of large non-metallic inclusions (>5μm) in small-sized bearings is low. However, maintaining such levels of material quality becomes more challenging in larger bearings and hence WTMBs may well have such inclusions present. Bearing manufacturing defects such as improper dimensional tolerances, unequal diameters of rolling elements and design flaws can also result in premature bearing failures (ZKL; Nabhan et al., 2015). Manufacturing defects can also potentially cause tight-fitted bearings or bearings with incorrect outer or inner diameter tolerances. A tight-fitted bearings exceeding its radial design clearance will experience excessive loads and overheating issues which both accelerate wear and fatigue (Barden). In the other direction, bearings fitted loosely between the outer raceway and housing will experience slippage and potentially fretting, wearing out the outer raceway surface (Barden). Finally, misalignment from improper assembly can result in overloading, vibration and overheating of bearings (He et al., 2012).

From the above descriptions it should be clear that most of these damage mechanisms are self perpetuating, generally leading to increasing amounts of similar damage. Furthermore, the presence of one can result in the initiation of others, for example: micro-pitting → spalling → abrasive wear.

## 7 Design standards

There are several ISO standards which underpin wind turbine bearing design specifications. ISO 76 (ISO, 2017a) deals with static load ratings of conventional rolling element bearings. This covers conventional design regarding the shape of rolling contact surfaces and assumes double row bearings to be symmetrical. ISO 76 states that it is not satisfactory in applications where:

- there is considerable truncation[6] of contact area between rolling elements and raceways

- application conditions cause deviations from a normal load distribution in the bearing. This can be due to preload, large clearance and surface treatments or coatings.

Bearings are generally designed to avoid truncation but occurrence can cause large reduction in MB life through acceleration of many of the above damage mechanisms. Since time-varying loads and large moments are present in the WTMB case, it is possible that these effects are present for wind turbines. ISO 281 (ISO, 2007) deals with dynamic load ratings and rated life calculations for conventional roller bearings where life is predominantly dependent on rolling contact fatigue. The same limitations apply as those noted above for ISO 76. In addition to this, the life calculations in ISO 281 do not account for the influence of wear, corrosion or electrical erosion[7] (ISO, 2007). ISO/TS 16281 (ISO, 2008) then allows for effects resulting from tilting, misalignment, clearances and internal load differences to be accounted for in rolling contact fatigue life calculations.

The above ISO standards form the basis for bearing design and certification in the turbine specific standards by the IEC and DNV-GL. Both the IEC (IEC, 2016, 2013) and GL (GL, 2010, 2012) standards for wind turbines require the use of ISOs 76 and 281 in combination with simulated loads across various operational cases. Additionally, both IEC and GL standards reference ISO/TS 16281. While in both cases the gearbox is given its own section, the MB is not considered specifically in either set of standards. Furthermore, MB design and rolling element selection is made based on fatigue life and static load ratings only. Design implications with respect to the likelihood of other possible damage mechanisms, i.e. those presented in the previous section, are not considered beyond potential adjustments to the modified life $L_{nm}$.

---

[6]Truncation is where element-raceway contact moves to the edge of the raceway, resulting in edge stresses, a smaller contact area and overall increased contact pressure.

[7]This being removal of material from bearing contact surfaces due to the presence of unintended electrical currents or voltages.

# 8    Fault Diagnosis and Prognosis

Rotating bearing monitoring is commonly done through the analysis of vibration signals using either vibration acceleration (VA) measurements or acoustic emissions (AE). These signals can then be analysed in either the time or frequency domain. Commonly, a set of so called *characteristic features* - such as signal RMS value, frequency component amplitudes or statistical moments - are extracted which are then used to relate the signal to previous operating data or known fault cases. When a rolling element passes a local defect on the race surface an impulse is generated which will then repeat as further passes are made. The frequencies of these events are dependent on bearing geometry, rotational speed and the location of the defect. Similarly, a defect on a bearing roller generate impulses as it moves around the bearing circumference which can be detected. The majority of VA and AE methods which have been developed work well in high speed applications. However, as discussed previously, WTMBs are components which operate at low-speeds and high-loads. This has the effect of making them less sensitive to vibrations as a result to defects or damage within the bearing. Therefore, many of the techniques which work well in high speed and moderately loaded cases have been found to be unreliable in this case. In addition, for the case of AE, the size of the MB structure means that signals are very susceptible to attenuation and hence information loss, reducing the usefulness of this data. Defect identification challenges resulting from the lower speed of the MB may be partially mitigated by recording longer sampling periods of vibration data. For example, a ten second data acquisition period has been found to be less successful in detecting low speed bearing defects than for high speed bearing/gear tooth defects (Carroll et al., 2019). This suggests that one cause of reduced detection rates for faults in low speed bearings may therefore be due to insufficient sampling periods. Some techniques have been proposed for MB fault detection which use vibration signals; as will be outlined, these techniques tend to also require additional measurements related to the turbine's operation.

In Qu et al. (2011) a MB fault diagnosis technique is proposed whereby stress waves measured in the MB are decomposed using wavelet analysis. It is suggested that faulty vs healthy cases can be classified according to their $D_3$ and $D_4$ signals. This work is based on a finite element model of a MB under the assumption that the bearing housing is rigid. Practicalities of how such measurements should be obtained in practise are not discussed. Further work, including real world testing and measurement technique consideration, would be necessary before this method became practicable. Zimroz et al. (2012) looks to diagnose a MB fault by processing signals of peak-to-peak vibration accelerations and turbine power. Real turbine data is used which covers operation with a faulty MB and operation after replacement with a non-faulty MB. Data from rated power operation and close to start-up are filtered out. The proposed method is then to perform a straight line fit to power vs peak-to-peak values, and it is shown that for the given fault example the resulting fit to 'good' data lies below that of 'bad' data. While this is indeed true for the fault example given, the fact that it is the only example makes it difficult to evaluate whether these results hold in general. Furthermore, for such a method to be usable it would be necessary to determine how different two fits need to be to signal that a fault is indeed present. Overlaps in the data from both cases make this task harder and a large number of datasets, including both healthy and faulty MBs, would be needed to properly assess its applicability and robustness. Ghane et al. (2016) propose the use of main shaft VA measurements to detect possible defects in the downwind MB of a DMB drivetrain. MB damage is modelled as a change in stiffness of the locating bearing unit in the axial direction. A

cumulative sum method is used to process acceleration signals and indicate when the probability of a change having happened is high enough to trigger a fault warning. A threshold is set which trades off between speed of detection and the likelihood of a false alarm. The data used in the study is obtained via simulation of a 5MW floating turbine in conjunction with a multibody gearbox model. In Hamadache and Lee (2016), a fault detection approach is presented based on shaft-speed measurements.

Vibration frequencies introduced to the system (by various types of bearing fault) are known to manifest themselves in the generator stator current. This in turn impacts generator torque and rotational speed. In steady state conditions, the impact of this can be modelled as introducing sinusoidal fluctuations in rotational speed, along with some noise. It is proposed that the detection of these sinusoidal components can warn of a fault being present, while the associated frequencies can indicate the location of said fault. It should be noted that this approach assumes steady-state operation and a fixed torque at the generator.

Absolute Value Principal Component Analysis is used as the detection method. Test cases are presented using simulated data with bearing faults represented by the aforementioned rotational speed variations, including a realistic wind speed input time history. The method behaves well for the example data, however, whether or not the various faults manifest themselves as assumed, while also being detectable from real system measurements, is an open question. Finally, although in test cases the method was found to work while in non-steady operation, given that steady-state operation is assumed, it is not clear whether

a similar performance would be seen across a turbine's whole operating envelope. Similar approaches to MB fault diagnosis have been proposed which also look to exploit the presence of these same vibration frequencies in the case of a MB fault. For DD bearings Teng et al. (2016) and Wang et al. (2017) do this by way of decomposing vibrations signals using a multiscale enveloping spectrogram and multiscale filtering spectrum respectively; Gong and Qiao (2013) develop an approach which looks for relevant frequencies appearing in stator current signals using a rotational-speed invariant PSD algorithm. These latter

three techniques explicitly account for varying rotational speeds during operation. Zhang (2018) documents an approach to MB fault prediction based on turbine SCADA measurements. An Artificial Neural Network (ANN) is used to determine a normal behaviour model of MB temperature at timestep $t$ as related to MB temperature at time $t-1$ and rotational speed, active power and ambient temperature at timesteps $t$ and $t-1$. Having trained the ANN on 6 months of normal operational data, deviations between predicted and measured outputs of MB temperature are used to warn of faults. False positives are avoided

by considering the percentage of time for which warnings are present over a week of operation, with an alarm threshold of 25%. Successful fault prediction on real turbine data is shown for 4 available fault cases from an operational wind farm. For the considered fault cases, the lead times on alarm activation are between 2.5 months and 3 days before failure.

The wind industry itself has become adept at detecting faults in WTMBs, with the most commonly used signals being VA (with a focus on ball-pass frequency) and MB temperatures (Romax-Technology; Foundation-AI and Ensemble-Energy).

Indications of MB faults 5-6 months prior to failure are achieved fairly consistently.

## 9 Discussion

This paper has given a broad overview of the existing theory and literature about, or relevant to, WTMBs. Aspects ranging from wind field structure all the way down to material degradation and damage have been considered in order to account for the

full range of interactions necessary for a complete picture of MB loading and lifetime. This also allows for cross-disciplinary understandings to be facilitated since, as has been shown, WTMBs must be considered as sitting at the interfaces connecting a number of disciplines.

The main conclusion which can be drawn from this literature survey is that much future work is required in order for WTMBs to be properly understood. It is clear that the WTMBs have received much less attention than other drivetrain components in terms of both design standards and research focus. This is understandable given the lack of information properly detailing MB failures in terms of frequency and damage modes. With respect to future efforts in this area, failure data concerning MBs (**D** in Figure 1) is therefore of the utmost importance. Knowledge of the types of damage occurring in the field are needed to drive research directions with respect to both numerical modelling and experimental analyses. Equally crucial to this topic is the development of a proper understanding and characterisation of MB loads, *i.e.* hub and LSS loading and subsequent load distributions about bearing circumferences (**A** to **C** in Figure 1). This includes wind field effects, controller interactions and design implications for time-varying loads. Understanding these factors will allow for a proper appraisal of current design standards (the underlying assumptions for which seem to be dubious at best in the MB case) and recommendations for improved design practises. A detailed understanding of how these various factors influence MB loads and lifetimes will also indicate potential solutions. For example, it could be the case that detrimental and damaging MB loads can be easily alleviated through controller design, equally, MB design considerations may in fact hold the key to lifetime extension for these components. Likely, a combination of approaches will be necessary. Finally, while a critical mass of existing MB research has taken place in the areas of detection and prognosis for MB faults, existing methods have been fairly generic and not specific to WTMBs. Therefore, a better understanding of load-damage relationships and prevalent failure modes for operating WTMBs could allow for more tailored approaches to failure detection and prognosis. This could well lead to improved results compared to the, already promising, work which has been done in this area to date.

## 10   Conclusion

This paper has reviewed existing literature and theory relevant to WTMBs in terms of design, operation, modelling, damage mechanisms and fault detection. The need for more research focusing on MBs in wind turbines was highlighted. Work in the literature relating to wind field structure and resultant loads at the WT hub, subsequently reacted at the MB, was presented. Common configurations and rolling elements for WTMBs were discussed with respect to both geared and DD turbines. The most common approaches to bearing modelling and analysis, falling into dynamic and quasi-static categories, were then outlined. After this, the tribology of rolling bearings and an overview of relevant damage mechanisms was presented. Finally, existing work in the literature which considers the diagnosis and prognosis of WTMB faults was summarised. The paper closed with a discussion of the most important next steps for research in this area, with key requirements being a need for detailed WTMB failure data and the development of a proper understanding of WTMB operational loading.

*Competing interests.* The authors declare that they have no conflicts of interest.

*Acknowledgements.* This work was funded by the EPSRC under grant number EP/R513349/1.

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
