# Peer review of "A review of wind turbine main-bearings: design, operation, modelling, damage mechanisms and fault detection"

_Wind Energy Science, 2019_

## Short Comment (SC1) · 28 May 2019

The article should probably state somewhere that it is largely limited to studying rolling element bearings. I say this because using plain bearing(s) as the main bearing in a wind turbine is also an active area of work, but they are not discussed here. The sections of the current article that are applicable to each is the applied loads. Another choice is to expand the scope of the paper and include this emerging plain bearing work.

Regarding the discussion in Section 1 on failure rates, another source for main bearing failure rates in the U.S. is Dan Brake of NextEra Energy's presentation -

Brake, D. (April 2013). "WTG SRB Main Bearing Failures." Presented at the 2013 UVIG Wind Turbine/Plant Operations & Maintenance Users Group Meeting. This source is not publicly-available though, so an alternate reference is from NREL at https://www.nrel.gov/docs/fy15osti/64311.pdf. A more recent, but less technical reference, is available at https://www.powermag.com/extending-turbine-life-to-meet-wind-powers-potential/.

In Section 3, I'm not sure I'd entirely agree with the statement that "The principal role performed by the MB is that of supporting the rotor while reacting non-torque loads, preventing them being transmitted further down the drivetrain." This depends on the type of the bearing (SRB, CRB, TRB) and the number of supports (three or four-point). The statement is true for a TRB in a 3-point drivetrain or a 4-point, but not an SRB in a 3-point. Here, the rotor moments are really reacted by corresponding support forces from the gearbox and mounts. This is backed up by the statement in Section 4.3 that "SRBs cannot support moment loads in either single or double-row configurations (Harris and Kotzalas, 2007)."

In Section 4, I suppose it could be added that GE is also essentially committing to DD technology for offshore machines with the Haliade and Haliade-X, moving up this mention from 4.2 into 4.

In Section 4.3, I would recommend to change "...WTMBs generally consist of two or more individual bearing units..." to "...WTMBs generally consist of two or more individual bearing rows..."

In Section 5.2, you could probably also mention Timken's Syber (https://www.timken.com/resources/high-performance-srb-technical-white-paper/)

In Section 5.3, I would say "Various forms of commercially-available FEM software are available..." distinguishing from the privately-held SKF, Schaeffler, and Timken software mentioned earlier.
In Section 6.1, I would say "...resulting in increased friction and wear."

In Section 8, it appears temperature monitoring is not mentioned until much later in the paragraph. This is a common and successful technique of main bearing fault diagnosis. I will also admit that I'm not entirely sure this section of the paper is needed. Although related to the main body of work, I would vote for expanding other portions of the review and eliminating this.

Generally speaking, this review does miss some important recent work such as at NTNU, RWTH-Aachen, and the bearing manufacturers. Several examples can be found at the Conference for Wind Power Drives in 2017 and 2019 or TORQUE 2018. So, in Section 9 I'm not quite sure I'd say "It is clear that the MB has been neglected in terms of...research focus". "Neglected" as a research focus may be a bit stronger than warranted.

---

## Author Comment (AC1) · 4 Jun 2019

Dear Jonathan,

First, on behalf of all the authors, I would like to thank you for taking the time to read our paper and provide valuable feedback.

With regards to rolling versus plain bearings - this is a good point and you are correct that we need to make sure this distinction is clear in the article. We will discuss whether to stick with a focus on rolling bearings, while indicating the existence emerging plain bearing use, or alternatively whether to expand the paper somewhat to include plain

bearings more fully.

With regards the discussion in Section 1 - thank you for pointing out these additional references which include information about main-bearing failures that we had not picked up on. We will make sure to include them in the revised manuscript.

With regards to Section 3 and the principal role of main-bearings - yes this is a fair point. The intent was to highlight that the main bearing is always supporting the rotor and contributing to the reaction of non-torque loads - however I agree that in its current form the sentence implies this role is exclusively performed by the main bearing. We will amend this for the revised manuscript.

For section 4 and GE and DD - good point, we will do as suggested.

For Section 4.3 and 'bearing units' vs 'bearing rows' - We take your point, however there is maybe some room for confusion here (for which we might need to reword a bit to avoid). A three point mount has a single bearing 'unit' which will commonly contain two bearing rows within it. A four point mount system has two bearing 'units' again with either one or two 'rows' within each of them. When discussing the WTMB in this section we are referring to the whole ensemble of either one or two entire bearings for the three and four point mount cases respectively. We will go away and consider how to make these distinctions as clear as possible, and will consider re-wording if necessary. Having explained our intended meaning I would be interested to hear your thoughts on whether a clarification will clear this up, or whether this ensemble description should maybe be re-thought.

For Section 5.2 - thank you, another helpful resource and reference. We'll add this in.

For Section 5.3 - good point, we'll add this in.

For Section 6.1 - This wording was intentional in just focusing on friction at this stage. Tribologically speaking the wear is a result of increased friction, but, only guaranteed if certain other factors relating to material properties are also present. The distinction is

somewhat academic but was felt to be important. We will however re-discuss this for the revised manuscript.

For Section 8 - I agree that having spoken to industry members it is clear that temperature monitoring has been allowing for successful fault diagnosis. However, we have not found published materials documenting this beyond what is presented in Section 8. This makes it difficult to include a more detailed discussion of its merits. We will have another look through the literature in case any new references for temp monitoring can be found. - in terms of removing Section 8 from the paper, from discussions we have had with both academics and industry members we are aware that, for quite a few parties, the diagnostics and prognostics for bearings is actually their principal interest when it comes to this topic area. As such, I don't believe such a review paper would be complete without these topics being included. We will of course still be happy to expand the other areas of the paper to include your suggested additions.

In terms of recent works - It is important that we find these references we may have missed. We actually finished and submitted this manuscript just before the TORQUE 2018 and some other conference proceedings became available and so there are indeed a few new items in those that we will add into the revised manuscript. We will also look through the other sources you have mentioned. In terms of whether main-bearings have been neglected or not - 'neglected' might be a bit strong, however, it does still seem to ring true when main-bearing research is compared to that of gearboxes and generators etc. However, we don't want to do a disservice to what good work has been going on and so we will revisit this terminology and perhaps dial it back a bit.

Thank you again for your valuable comments and if you have any further questions or comments I'll be most happy to respond to them.

Best regards,

Edward Hart (Corresponding author)

---

## Referee Comment (RC1) · Anonymous Referee #1 · 20 Jun 2019

This is a review paper on the design, operation, wear and fault detection of rolling element wind tubine main bearing systems. The review is well written with a clear structure throughout the paper. The authors should consider the following minor corrections/clarifications/additions to the paper.

1. Section 2 is short. The authors should consider adding a little bit more detail on the fundamental differences between geared and direct drive wind turbine drivetrains and perhaps a figure or some text that quanitifies the past and predicted future growth of rated wind turbine output and rotor size/weight over time. 2. In section 3.2, the text in the first few lines on page 7 concerning inputting expressions for blade moments in the

form of mean plus fluctuating components into Equations 5 & 6 resulting in 'the mean values cancelling' is not clear. The authors should consider adding some further steps in the analysis being described here. 3. Section 3.2.1 lines 14 & 15 It is stated that from equations 5 & 6 it follows that 'blade root bending moments with range Mrange result in hub moment fluctuations of 1.5Mrange.'. Again, it is not immediatley clear how this conclusions follows from these two equations. A more complete explanation is needed. 4. The authors should consider using 'gearbox end' and 'rotor end' when describing the low speed shaft rather than 'downstream end'. 5. Consider starting a new sentence in Section 4.1, line 2 on page 12 '....(Bergua et al, 2014). While this may be the case .....'. 6. Section 4.2 line 13 on page 12.'including having bearings in the air-gap diameter'. What does this mean? Should 'diameter' be 'clearance' or is something different being described here? 7. Section 7 page 21 line 8. The text in this line includes 'electrical erosion' the authors should state now this differs from wear & corrosion (already listed) of electrical components?

In summary this is a good review paper that in my opinion is worthy of publication in the journal subject to consideration of the minor points listed above.

---

## Author Comment (AC2) · 3 Jul 2019

Dear Sir/Madam,

First, we would like to thank you for taking the time to review our paper and providing valuable feedback and comments which will help to improve this work.

Response to specific comments:

1. This is a good point and we agree that additional information and a figure depicting growth in turbine power and rotor weight would be a valuable addition.

2. In the revised manuscript we will add additional steps to ensure the logic pertaining

to Equations 5 & 6 and the subsequent cancelling of the mean term is clear.

3. As above, we will add additional information here to ensure clarity.

4. We agree that the use of 'rotor end' and 'gearbox end' would make this part clearer and so will make this change in the revised manuscript.

5. We agree with this suggested grammatical change.

6. You are correct that 'diameter' here does mean 'clearance' inside of the generator. We had followed the wording of previous authors in describing this configuration but for the sake of clarity we will review the description and update it.

7. We will add additional information here to clarify this difference. Electrical erosion here refers to stray currents present in a bearing using rollers to go to earth, this can initiate or exacerbate pitting and spalling on the bearing itself. Hence, rather than being the erosion of electrical components, we are instead referring to electrical currents causing erosion of mechanical components.

Thank you again for your input, we will endeavour to improve this work by updating the paper according to the excellent points you have made.

Best,

Edward Hart (Corresponding author)

---

## Referee Comment (RC2) · Babak Eftekharnejad (Referee) · 14 Oct 2019

In general the quality of the review is not adequate in many aspects and in some cases lacks accuracy and up to date information on state of art. It is recommended that the authors to revisit the content of the this article and ensure review is based on the most update information referencing to recent advancement in design ,diagnostic and manufacturing of the main bearing. As such find below some specific comments :

 ́c Section 3.2 : it is not clear how the equation (6) and (5) are derived given the illustration in figure 2. It is more appropriate to use the standard GL coordinate system which is common in this industry and then explain the hub loading .  ́c Section 4.1:

The most important drawbacks of using integrated drivetrain (figure 6C) is the lack of ability to exchange main bearing independent of nacelle in event of failure ; i.e if the main bearing fails then the entire nacelle needs to be exchange hence the high cost of repair particularly in offshore environment. It is recommended that the author carry out through an accurate comparison between different platform design with view on maintainability . • Section 6.1 paragraph 10 : Please note , the main bearing on modern DD turbine ,e.g SGRE D6 platform , is grease lubricated and employs quite advance and sophisticated grease pumping system . Also at least for the past 7-8 years no manual greasing of main bearings is being conducted but using auto –greaser or automatic grease pumps. Please revisit the statement in this section and ensure it reflects the current practice . • Section 6.2 : It is recommended that the author to review the damage related to material and manufacturing of the bearings ;i.e sub case and sub-surface cracks. This is very common type of main bearing failure particularly in larger offshore machines. • Section 8 paragraph 20 : AE is not a common method to monitor bearing or at least not main bearing in wind industry. The main bearing is a large structure and AE signals are very susceptible to attenuation particularly across large transmission path and hence measuring signals from the bearing casing will not yield into meaningful information. As a general comments main bearing failures are among the most straightforward and is easily detectable through motioning of ball pass frequency across the spectra . This is easily detected through available commercial condition monitoring system in turbine and generally 5-7 months in advance of turbine shut down .

---

## Author Comment (AC3) · 21 Oct 2019

Dear Babak,

We would like to start by thanking you for taking the time to review our article and make very valuable comments and suggestions. Your points highlight some of the differences between industry practises and existing literature, the identification of which will make this paper all the more relevant to wind energy research as well as the wind industry itself.

We reply to each of your comments specifically below.

Regarding Section 3.2: We agree that more detail regarding this derivation would improve clarity here, these relevant steps will therefore be added into the revised manuscript. In terms of the reference frame used we have used the reference frame of the text (The Wind Energy Handbook) from which this description is taken, showing how fluctuations in hub loading can be expressed as fluctuations about a mean with deterministic and stochastic components. The current version of the paper does this consistently with the original source and we feel it could create more confusion to deviate from that original description. Therefore on this point we would like to keep the existing reference frame, although we are happy for the editor to have the final say here.

Regarding Section 4.1: You raise the very important point of maintainability as associated to the various drivetrain designs. We agree that a discussion of this should be included and will add this into the revised manuscript. Published literature that discusses these aspects in detail is somewhat thin on the ground but we have found some good references which will allow us to bring it into the conversation.

Regarding Section 6.1: Thank you for raising this point. Again this highlights the lack of discussion of these maintenance approaches in the academic literature. We will revise this sentence as you have suggested and have also found sources which describe these systems which will also be included.

Regarding Section 6.2: An excellent suggestion. We will expand this section to include failures related to materials and manufacturing including influences of steel cleanliness and heat treatments, inclusion microstructures etc.

Regarding Section 8: Just to clarify, we had not been claiming that AE is used for wind turbine main-bearings. This part of Section 8 was outlining the common techniques (vibration and AE) and discussing the difficulties both can have in lower speed applications and for MBs in particular. The insight into AE attenuation is very helpful and we will add this to that section. Section 8 is the main one for which there is a large

discrepancy between industry practise and the academic literature. You are quite right of course about the available CMS softwares, interestingly there is very little cross-over between them and the current literature (which this section covers). Therefore, it seem this discrepancy needs to be discussed directly in the paper to point out that academic research is not aligned well to what is happening in the field. We will therefore discuss some of the existing products and the signals on which they make predictions. You specifically mention the ball-pass frequency as a detection method. We assume that this is measured using processed vibration data, and so does align with the techniques we discuss in this section, but we'll make sure to mention it more specifically. From discussions with industry partners we have found a general opinion that for younger turbines the detection of these faults is relatively easy as you have mentioned, but that as a fleet ages it becomes more difficult as false-positives start to creep in. In this latter situation a broader knowledge base and range of techniques driven by academic research may well give valuable detection improvements, and so we feel there is value to this area being given research focus and hence being included in the paper here. We will try and bring out some of the nuances of this discussion when revising the manuscript, although as ever there is limited literature discussing this.

Again, we thank you for the time and effort you have put into appraising our paper. We believe that your comments have been most valuable and will significantly improve this work.

If you have any further questions or points which you would like clarified then we'll be happy to respond.

Many thanks on behalf of all co-authors,

Edward Hart (corresponding author)

---

## Author Comment (AC4) · 21 Oct 2019

Dear Athanasios,

Thank you for managing the review process for this paper. We appreciate the excellent points made by the 3 reviewers who have commented on this work.

Overall we feel these reviews indicate that the paper is of a high quality and worthy of publication with a majority of points either being minor adjustments or valuable suggestions for additional references or discussion points. In particular we feel that the opening remarks made by the third reviewer (who again made excellent and helpful

points) are not really reflective of the points they then go on to make. However, their points do highlight that there are some marked differences between industry practises and what is being focused on in academic research and we will seek to highlight these differences when revising the manuscript in order to stimulate better alignment in the future. This disparity between academia and industry for wind turbine main-bearings further motivates the need for a review paper which captures the current state of knowledge for this important component and, thanks to this review process, this paper will be extended to capture and discuss that disparity, fulfilling that need.

Since the submission of this manuscript for review there have been a number of papers published in journals and conference proceedings dealing with main-bearings in wind turbines. This new work, along with the additional references suggested by reviewers, will be included in the revised manuscript.

In conclusion we believe that this is an important piece of work, with the suggested additions provided by reviewers comments helping to improve it and ensure it properly captures the current state of main-bearing technology.

Many thanks on behalf of all co-authors,

Edward Hart (corresponding author)

---

## Author Response (AR1)

Comments and Responses for WES-2019-25 (revised manuscript with changes highlighted appended below)

The comments from each of the three reviewers are listed below followed by our response, edits to the revised manuscript and the page and line numbers where the changes can be found in the tracked version of the manuscript.

Jon Keller

Comment: The article should probably state somewhere that it is largely limited to studying rolling element bearings. I say this because using plain bearing(s) as the main bearing in a wind turbine is also an active area of work, but they are not discussed here. The sections of the current article that are applicable to each is the applied loads. Another choice is to expand the scope of the paper and include this emerging plain bearing work.

Response: This clarification has been added as suggested. Page 3, line 5.

Comment: In Section 3, I'm not sure I'd entirely agree with the statement that "The principal role performed by the MB is that of supporting the rotor while reacting non-torque loads, preventing them being transmitted further down the drivetrain." This depends on the type of the bearing (SRB, CRB, TRB) and the number of supports (three or four-point). The statement is true for a TRB in a 3-point drivetrain or a 4-point, but not an SRB in a 3-point. Here, the rotor moments are really reacted by corresponding support forces from the gearbox and mounts. This is backed up by the statement in Section 4.3 that "SRBs cannot support moment loads in either single or double-row configurations (Harris and Kotzalas, 2007)."

Response: The sentence has been amended to "The principal role performed by the MB is that of supporting the rotor while reacting non-torque loads either independently, preventing them being transmitted further down the drivetrain, or in combination with the gearbox and mounts". Page 3, line 28.

Comment: In Section 4, I suppose it could be added that GE is also essentially committing to DD technology for offshore machines with the Haliade and Haliade-X, moving up this mention from 4.2 into 4.

Response: This has been added as suggested. Page 12, line 2.

Comment: In Section 4.3, I would recommend to change "...WTMBs generally consist of two or more individual bearing units..." to "...WTMBs generally consist of two or more individual bearing rows..."

Response: On reflection we have decided that this sentence is a bit misleading since, as pointed out by the reviewer, different people will interpret it in different ways. Overall it is actually not a crucial sentence to leave in and so we have simply removed it instead. This bit now reads "Having outlined common roller types and their typical configurations, we will now detail which bearings are used in the various drivetrain setups." This avoids the ambiguity and potential confusion of the original statement. Page 15, line 8.

Comment: In Section 5.2, you could probably also mention Timken's Syber (https://www.timken.com/resources/high-performance-srb-technical-white-paper/)

Response: This additional system has been added to the discussion as suggested. Page 18, line 13.

==Comment==: In Section 5.3, I would say "Various forms of commercially-available FEM software are available..." distinguishing from the privately-held SKF, Schaeffler, and Timken software mentioned earlier.

==Response:== Have changed this sentence as suggested. Page 19, line 19.

==Comment:== In Section 6.1, I would say "...resulting in increased friction and wear."

==Response:== This sentence has been amended as suggested. Page 19, line 32.

==Comment:== In Section 8, it appears temperature monitoring is not mentioned until much later in the paragraph. This is a common and successful technique of main bearing fault diagnosis. I will also admit that I'm not entirely sure this section of the paper is needed. Although related to the main body of work, I would vote for expanding other portions of the review and eliminating this.

==Response:== An additional paragraph has been added which describes the successes in industry of MB fault detection using temperature and ball-pass frequencies. In terms of whether or not this section should be included we feel it is an important area concerning the main bearing and hence important to keep as part of the review. Page 26, line 3.

==Comment:== Generally speaking, this review does miss some important recent work such as at NTNU, RWTH-Aachen, and the bearing manufacturers. Several examples can be found at the Conference for Wind Power Drives in 2017 and 2019 or TORQUE 2018. So, in Section 9 I'm not quite sure I'd say "It is clear that the MB has been neglected in terms of...research focus". "Neglected" as a research focus may be a bit stronger than warranted.

==Response:== The suggested additional sources for references have proved very useful and these additional works have now been included in the paper. We have also changed the perhaps overly strong sentence "It is clear that the MB has been neglected in terms of...research focus" to instead read "WTMBs have received much less attention in the research literature than other drivetrain components". Page 26, line 13. With respect to the missing work from the literature we have been through the suggested sources and added to what is captured in this paper. This includes the additional references Kasiri et al. (2019), Schröder et al. (2019), Cardaun et al. (2019), Kock et al. (2019) and Reisch et al. (2018).

**Anonymous referee #1**

==Comment:== Section 2 is short. The authors should consider adding a little bit more detail on the fundamental differences between geared and direct drive wind turbine drivetrains and perhaps a figure or some text that quantifies the past and predicted future growth of rated wind turbine output and rotor size/weight over time.

==Response:== Detail comparing geared and direct drive has been added which reads "Drivetrain choices have proved an area for which a consensus on optimal design has not yet been reached, the main split relevant to the current work being between geared and direct-drive (DD) machines. Geared machines use a gearbox to step up the slow rotational speed of the turbine rotor to the fast (around 1800 rpm) rotational speed needed to generate electricity using conventional generator technology. The gearbox is a complex piece of machinery which has traditionally been seen as a fault prone component that reduces overall reliability. In response to this, DD machines were developed. These remove the gearbox entirely and, in order to be able to generate at slow rotational speeds, have increased generator diameters and operating torques. But, this switch to low speed generation

comes with associated increases in cost, size and weight and so the optimal solution is still an open question, especially considering increasing reliabilities of geared machines." With regards to the suggested figure this has also been added and is Figure 2 in the revised manuscript, it details average rotor diameters and power ratings for turbines in the US and Germany from 2006-2018. Page 3, line 19 and Figure 2 on page 4.

Comment: In section 3.2, the text in the first few lines on page 7 concerning inputting expressions for blade moments in the Discussion paper form of mean plus fluctuating components into Equations 5 & 6 resulting in 'the mean values cancelling' is not clear. The authors should consider adding some further steps in the analysis being described here.

Response: More detail has been added here to aid clarity. Page 8, line 4.

Comment: Section 3.2.1 lines 14 & 15 It is stated that from equations 5 & 6 it follows that 'blade root bending moments with range Mrange result in hub moment fluctuations of 1.5Mrange.'. Again, it is not immediatley clear how this conclusions follows from these two equations. A more complete explanation is needed.

Response: More detail has been added here to aid clarity. Page 8, line 19.

Comment: The authors should consider using 'gearbox end' and 'rotor end' when describing the low speed shaft rather than 'downstream end'.

Response: This phrasing has been changed to that suggested by the reviewer throughout the paper.

Comment: Consider starting a new sentence in Section 4.1, line 2 on page 12 '....(Bergua et al, 2014). While this may be the case .....'.

Response: Sentence amended as suggested.

Comment: Section 4.2 line 13 on page 12.'including having bearings in the air-gap diameter'. What does this mean? Should 'diameter' be 'clearance' or is something different being described here?

Response: 'diameter' replaced with 'clearance' here as suggested. Page 13, line 26.

Comment: Section 7 page 21 line 8. The text in this line includes 'electrical erosion' the authors should state now this differs from wear & corrosion (already listed) of electrical components?

Response: Footnote added to clarify this difference. Footnote reads "This being removal of material from bearing contact surfaces due to the presence of unintended electrical currents or voltages." Page 23, footnote 7.

Babak Eftekharnejad

Comment: Section 3.2 : it is not clear how the equation (6) and (5) are derived given the ´ illustration in figure 2. It is more appropriate to use the standard GL coordinate system which is common in this industry and then explain the hub loading .

Response: More detail has been added here to aid clarity in the derivation. A footnote addressing the reference frame and citing the more standard GL one is also added. Page 7, line 12 and Page 7, footnote 2.

Comment: Section 4.1: The most important drawbacks of using integrated drivetrain (figure 6C) is the lack of ability to exchange main bearing independent of nacelle in event of failure ; i.e if the main bearing fails then the entire nacelle needs to be exchange hence the high cost of repair particularly

in offshore environment. It is recommended that the author carry out through an accurate comparison between different platform design with view on maintainability .

Response: The issues relating to the integrated drivetrain are now mentioned when it is introduced in the text. In terms of a 'thorough comparison between different platforms design with regards to maintainability' this is tricky as there is no real data in the public domain that allows for such a comparison in much detail. We have therefore instead brought out the discussion by considering the benefits of modularity versus a more compact integrated approach in order to bring out the importance of maintainability as suggested by the reviewer. To this end the following paragraph has been added: "Maintainability was mentioned above as being a key driver for drivetrain selection at the design stage. Section 4.1 contained a clear example of this with respect to fully integrated drivetrains requiring the entire nacelle to be exchanged in the case of a main bearing failure. This is a more extreme example but it highlights the point that the maintainability of a given system will play an important role in determining its overall viability. From a maintenance point of view, more modular concepts tend to allow for easier component accessibility and exchange. Partially-integrated designs have also been introduced as a way of trying to secure the benefits of both design philosophies (Gasch and Twele, 2012)." Page 13, line 7 and Page 13, line 29.

Comment: Section 6.1 paragraph 10 : Please note , the main bearing on modern DD turbine ,e.g SGRE D6 platform , is grease lubricated and employs quite advance and sophisticated grease pumping system . Also at least for the past 7-8 years no manual greasing of main bearings is being conducted but using auto –greaser or automatic grease pumps. Please revisit the statement in this section and ensure it reflects the current practice .

Response: This statement has been revised as suggested. Page 20, line 22.

Comment: Section 6.2 : It is recommended that the author to review the damage related to material and manufacturing of the bearings ; i.e sub case and sub-surface cracks. This is very common type of main bearing failure particularly in larger offshore machines.

Response: This was an excellent suggestion on the part of the reviewer. A new section has been added entitled '*Damage related to materials and manufacturing*' which is section 6.3 in the updated manuscript. This section reviews this topic and describes how these material, design and manufacturing faults can lead to premature bearing failures. Page 22, line 27.

Comment: Section 8 paragraph 20 : AE is not a common method to monitor bearing or at least not main bearing in wind industry. The main bearing is a large structure and AE signals are very susceptible to attenuation particularly across large transmission path and hence measuring signals from the bearing casing will not yield into meaningful information. As a general comments main bearing failures are among the most straightforward and is easily detectable through motioning of ball pass frequency across the spectra. This is easily detected through available commercial condition monitoring system in turbine and generally 5-7 months in advance of turbine shut down.

Response: The point about AE attenuation has been added as follows '…for the case of AE, the size of the MB structure means that signals are very susceptible to attenuation and hence information loss, reducing the usefulness of this data'. We have also provided more details about the successful use of ball pass frequency and temperature monitoring in industry to detect main-bearing faults with the following paragraph: "The wind industry itself has become fairly 
[revised manuscript text omitted]